# Nitric Oxide-Mediated Enhancement and Reversal of Resistance of Anticancer Therapies

**DOI:** 10.3390/antiox8090407

**Published:** 2019-09-17

**Authors:** Emily Hays, Benjamin Bonavida

**Affiliations:** Department of Microbiology, Immunology and Molecular Genetics, David Geffen School of Medicine, University of California Los Angeles, CA 90095, USA; emilyhays@g.ucla.edu

**Keywords:** nitric oxide, sensitization, chemotherapy, immunotherapy, cell signaling, targeted therapy

## Abstract

In the last decade, immune therapies against human cancers have emerged as a very effective therapeutic strategy in the treatment of various cancers, some of which are resistant to current therapies. Although the clinical responses achieved with many therapeutic strategies were significant in a subset of patients, another subset remained unresponsive initially, or became resistant to further therapies. Hence, there is a need to develop novel approaches to treat those unresponsive patients. Several investigations have been reported to explain the underlying mechanisms of immune resistance, including the anti-proliferative and anti-apoptotic pathways and, in addition, the increased expression of the transcription factor Yin-Yang 1 (YY1) and the programmed death ligand 1 (PD-L1). We have reported that YY1 leads to immune resistance through increasing HIF-1α accumulation and PD-L1 expression. These mechanisms inhibit the ability of the cytotoxic T-lymphocytes to mediate their cytotoxic functions via the inhibitory signal delivered by the PD-L1 on tumor cells to the PD-1 receptor on cytotoxic T-cells. Thus, means to override these resistance mechanisms are needed to sensitize the tumor cells to both cell killing and inhibition of tumor progression. Treatment with nitric oxide (NO) donors has been shown to sensitize many types of tumors to chemotherapy, immunotherapy, and radiotherapy. Treatment of cancer cell lines with NO donors has resulted in the inhibition of cancer cell activities via, in part, the inhibition of YY1 and PD-L1. The NO-mediated inhibition of YY1 was the result of both the inhibition of the upstream NF-κB pathway as well as the S-nitrosylation of YY1, leading to both the downregulation of YY1 expression as well as the inhibition of YY1-DNA binding activity, respectively. Also, treatment with NO donors induced the inhibition of YY1 and resulted in the inhibition of PD-L1 expression. Based on the above findings, we propose that treatment of tumor cells with the combination of NO donors, at optimal noncytotoxic doses, and anti-tumor cytotoxic effector cells or other conventional therapies will result in a synergistic anticancer activity and tumor regression.

## 1. Introduction

### 1.1. Cancer and Conventional Therapies

Cancer cells proliferate and survive by escaping the host’s regulatory systems through various mutations. These mutations lead to uncontrolled cell growth, tumorigenesis, metastasis, and death of the host. Different cancers have different types of mutations and cause disease through manipulating different cellular pathways and their environment. Thus, cancer therapies have been developed for different cancers that attempt to kill the cells through various mechanisms. One of the most common therapies is chemotherapy, which kills cells by damaging DNA and inhibiting mitosis [1]. Chemotherapy has led to significant clinical responses in many cancers, including increased cytotoxicity and prolonged overall and progression-free survivals, such as colorectal cancer [2], non-small-cell lung cancer [3], pancreatic cancer [4], and many others. Various combination chemotherapies have also shown to be more effective compared to single agent therapies in treating several cancers, such as advanced non-small-cell lung cancer [5] and metastatic breast cancer [6]. Another common method of treatment is radiotherapy, which uses high energy rays to kill cancer cells in a localized area [7]. Radiotherapy is often combined with other treatments and has been shown to be effective in treating cancers, such as rectal cancer, in combination with surgery [8] and chemotherapy [9]. Radiotherapy has also been shown to be effective in combination with immunotherapy in preclinical and some clinical studies because of its potential to change the tumor microenvironment and stimulate immune responses [10,11,12].

Immunotherapy aims to enhance the host’s innate, antibody, and cell-mediated immune attacks on cancer cells [13,14]. Remarkable progress has been made in recent years in cancer immunology and immunotherapies. Several recent therapeutic strategies, such as novel stimulator of interferon genes (STING) agonists, have been developed to enhance the host’s innate immune attack on cancer cells [15]. Helper-like innate lymphoid cells have also emerged as a new target for immunotherapies because of their ability to infiltrate the tumor microenvironment [16,17]. Targeted therapies, such as monoclonal antibodies, T-cell mediated therapies, and small molecule inhibitors aim to inhibit cancer cell growth, increase cell death, and restrict the spread of cancer. These targeted therapies target specific cancer proteins or molecular pathways and are preferred because they minimize the death of normal cells and specifically target cancer cells. Among the most notable targeted immunotherapies are checkpoint inhibitor therapies. Thus, antibodies that block the cytotoxic T-lymphocyte-associated protein 4 (CTLA-4), the programmed cell death receptor 1 (PD-1), or the programmed cell death ligand 1 (PD-L1) have been successful in enhancing the host’s attack on various tumor types, including melanoma, lung, bladder, and many others [14]. Cell-based immunotherapy is another targeted therapy approach. Because low numbers of tumor infiltrating lymphocytes (TILs) are correlated with poor survival in some cancers, cell-based immunotherapy aims to increase this number by isolating TILs from a patient’s specimen, expanding them in vitro, and re-infusing them back into the patient. However, this approach requires tumors with many antitumor T cells, which is uncommon in most tumors, and the process is difficult, labor intensive, and time consuming [18]. Thus, more advanced approaches have been developed in recent years, including adoptive cell transfer (ACT). In this approach, a patient’s T-cells are genetically modified with receptors for specific cancer antigens. T-cells can be modified with either T-cell Receptors (TCRs) or Chimeric Antigen Receptors (CARs). Since TCRs can only recognize antigens when presented on Major Histocompatibility Complexes (MHCs), the TCR approach is limited because many cancers downregulate the expression of MHCs on the surface of the cell. Additionally, TCRs can recognize small peptide epitopes and may cross-react with self-antigens [18]. However, CAR T-cell therapy uses chimeric proteins, which link antibodies that target tumor cell surface antigens to intracellular signaling receptors for TCRs. Both Phase I and Phase II clinical trials of an anti-CD19 CAR T-cell therapy, axicabtagene ciloleucel (axi-cel), showed efficacy in patients with B-cell lymphomas. In the phase II trial, the objective response rate was 82%, and the complete response rate was 54% [19]. Axi-cel was FDA-approved in 2017 for use in patients with large B-cell lymphoma and produced durable responses in most patients; however, some patients experienced cytokine release syndrome and other undesired side effects [20].

Small molecule inhibitors are another type of targeted cancer therapy. Various small molecules have been used to constrain cancer cell growth and survival. Their mechanisms of action can vary from inhibiting growth pathways to targeting apoptotic regulators, inhibiting proteins to reactivate p53 function, or targeting proteins, such as Hsp90, that promote malignant transformation. Small molecule inhibitors, such as these, have shown significant antitumor effects, including the increased apoptosis of tumor cells, and improved clinical outcomes in various types of cancer [21,22,23]. Small molecule inhibitors can also sensitize tumor cells to other therapies. Selective small molecules, such as DNA inhibitors, were found to sensitize cancer cells to chemotherapy [24]. Combinations of small molecule inhibitors can also sensitize radiation therapy-resistant cancer cell lines [25].

### 1.2. Resistance to Various Therapies and Mechanisms

Although many patients initially respond to current therapies, the response of many patients is not long-lasting. Additionally, a subset of patients is intrinsically resistant, and many of the initially responsive patients relapse and acquire resistance to further therapy. Many chemotherapies, radiotherapies, and immunotherapies often lead to drug resistance because of the selective pressure they assert on cells. Cancer cells acquire chemo-drug resistance by various mechanisms, such as by mutations or amplification of the drug’s target enzyme, overexpression of drug transporters, or mutations in cell-death pathways [26]. Radiation therapy can lead to resistance due to changes in DNA repair pathway utilization, DNA damage checkpoint activation, and energy metabolism [25]. Antibody immunotherapies can also lead to drug resistance. One study found that up to 60% of patients treated with the anti-PD-1 therapy developed resistance [27]. Another study has suggested that resistance can develop to CAR-T-cell therapy in B-cell acute lymphoblastic leukemia (B-ALL) due to epitope loss under therapy pressure [28]. Resistance can also be acquired to small molecule inhibitors such as the inhibitor of the epidermal growth factor receptor (EGFR) in malignant gliomas and lung cancer [29,30] or androgen receptor inhibitors in prostate cancer [31].

Several recent studies have proposed mechanisms for how resistance is acquired. The gut microbiome has been shown to be involved in the response to immunotherapies. A recent study showed that antibiotic consumption was associated with a poor response to the immunotherapeutic PD-1 blockade [32]. They also showed that nonresponding patients to the anti-PD-1 therapy were deficient in the bacterium *Akkermansia muciniphila,* and that oral supplementation of the bacteria to mice reversed resistance to immunotherapy [32]. Another study found that responding versus nonresponding patients to anti-PD-1 immunotherapy have significant differences in the bacterial composition of their gut microbiome [33]. Other factors have been shown to be involved in acquiring resistance to immunotherapies, such as the drug efflux transporter and other membrane drug transporters that shuttle drugs across cell membranes, protecting the cell from the accumulation of toxic drugs [34]. The transcription factor, YY1, has also been shown to regulate immune resistance by modulating the expression of PD-L1 in cancer cells through several crosstalk pathways [35]. The inhibition of YY1 sensitizes tumor cells to apoptosis [36] and may be a potential therapeutic target for overcoming immune resistance. Others have reviewed several other mechanisms of resistance. A summary of mechanisms of immune resistance is shown in Table 1.

### 1.3. Reversal of Resistance

Because of the rise of resistance to immunotherapies and other cancer therapies, combination therapies have been explored to treat cancers that do not respond to single therapies alone. For example, the combined use of checkpoint inhibitors for CTLA-4 and PD-1 can improve the clinical response in metastatic melanoma patients up to 60%; however, there is an increased frequency of toxicities [49]. Radiotherapies can also enhance immunotherapies. Radiation of tumor cells was shown to up-regulate Fas and enhance the cytotoxic T-lymphocyte (CTL) lytic activity and adoptive immunotherapy [50]. Small case studies have shown that patients with mucosal melanoma of the lower genital tract respond favorably to the combined treatment of a CTLA-4 antibody and radiation therapy [51]. Combination therapy of the CTLA-4 antibody, ipilimumab, and radiation therapy was also well-tolerated and effective in patients with stage IV melanoma without any unexpected toxicities [52]. Combined cytokine therapy with vaccines may also improve the immune response against tumors. In patients with stage IV or locally advanced stage III cutaneous melanoma, the combined use of interleukin-2 (IL-2) therapy and the gp100 peptide vaccine resulted in an improved response rate and progression-free survival rate compared to the use of IL-2 therapy alone [53]. Chemotherapy can also sensitize tumors to checkpoint blockade therapies. In a phase II study, a CTLA-4 antibody used in combination with paclitaxel and carboplatin chemotherapies improved the progression-free survival in non-small-cell lung cancer [54]. In another study done on tumors that lack CD8+ T cells, chemotherapy enhanced tumor T-cell infiltration, sensitized tumors to checkpoint inhibition therapies, and controlled cancer growth [55].

There have been many other studies and reviews that have shown how resistant tumor cells can be sensitized to immunotherapies and chemotherapies. Briefly, reactive oxygen species and their inducers have been found to sensitize tumor cells to apoptosis (Table 2). Small molecules, such as nitric oxide donors, can also sensitize resistant tumor cells to immunotherapy, chemotherapy [56], and radiation therapy [57]. Thus, nitric oxide donors and other agents that can increase the localized production of reactive oxygen species may be effective for use in combination therapies and overcoming drug resistance. The exact mechanisms of sensitization will be discussed later in this chapter.

Because of the abundance of studies showing that NO donors and other reactive oxygen species can sensitize cells to apoptosis, the combination of NO donors with cancer immunotherapies and chemotherapies may enhance the treatment of drug-resistant tumor cells. In the following sections, we will discuss the current progress of research on the role of nitric oxide in cancer and the immune system, the various therapies that have been developed, and the future of nitric oxide-based cancer therapies.

## 2. Nitric Oxide and Cancer

### 2.1. Introduction

Nitric oxide (NO) is a water soluble, free radical gas involved in many biological processes, such as vasodilation, neurotransmission, macrophage-mediated immunity, and anti-inflammatory responses [65,66]. Nitric oxide synthases (NOSs) catalyze the reaction between L-arginine and oxygen to produce NO and L-citrulline [67]. These nitric oxide synthases consist of the inducible nitric oxide synthase (iNOS), endothelial NOS (eNOS), and neuronal NOS (nNOS). NO has become a molecule of interest in cancer research because of the many studies that have found it to be implicated in various cancer processes, such as angiogenesis, apoptosis, cell cycle, invasion, and metastasis [68]. Some studies have shown that NO, at high levels, can generate DNA damage and promote cytotoxic effects and apoptosis and, at intermediate levels, can promote invasiveness, cytoprotection, and apoptosis [69]. Consequently, the role of NO in cancer is highly controversial because of the contrasting results of many studies.

### 2.2. Dual Role of NO in Cancer Biology

NO has been shown to have either tumor-inhibiting or tumor-promoting effects, depending on the type of cancer, the tumor microenvironment, the type of NO synthase, and several other factors.

Several studies have detected elevated expression and/or activity of NO synthases in cells from various human cancers, suggesting a correlative relationship between NO and the progression of cancer. Some of these cancers include breast cancer [70,71,72], gynecological cancer [73], head and neck cancer [74], colon cancer [75], prostate carcinoma [76], bladder cancer [77], gastrointestinal cancer [78], melanoma [79], and pancreatic cancer [80].

There are many mechanisms by which NO has been shown to contribute to the progression and aggressiveness of certain cancers. The exposure of NO donors has been shown to increase DNA synthesis, cell proliferation, and migration of endothelial cells in human and bovine endothelial cells [81]. NO can also contribute to cancer growth by promoting Vascular Endothelial Growth Factor (VEGF)-induced angiogenesis [82]. NO contributes to proangiogenic pathways by promoting cell growth through the activation of endothelial-constitutive NO synthase (ec-NOS), cyclic GMP, mitogen activated kinase (MAPK), and fibroblast growth factor 2 (FGF-2) [83]. NO derived from eNOS promotes angiogenesis and tumor progression through the maintenance of blood flow, induction of vascular hyperpermeability, and reduction of leukocyte–endothelial interactions [84].

Additionally, NO can inhibit apoptosis by downregulating proteins involved in the apoptotic signal transduction pathways [85]. NO acts as an epigenetic regulator of oncogenesis by causing histone post-translational modifications through direct inhibition of the catalytic activity of the JmjC-domain that contains histone demethylases [86]. These histone post-translational modifications can cause changes in the expression of oncogenic genes, such as *Bcl-2*, and apoptotic regulators, such as *Bax*, leading to oncogenesis [86,87]. NO can induce several pro-oncogenic pathways, such as the PI3K/Akt/mTOR, TGFβ, and ERK signaling pathways [88]. The induction of the mTOR pathway by iNOS has been shown to promote proliferation of human melanoma by the nitrosylation of the tuberous sclerosis complex-2 (TSC2) [89]. NO also regulates proinflammatory mediators, resulting in the oncogenic transformation of cells [90].

Contrastingly, NO has been shown to have several antitumorigenic effects. For instance, NO has an antitumorigenic effect of contributing to macrophage-induced cytotoxicity [91]. It is believed to have a cytotoxic effect because of its ability to sequester iron into iron-nitrosyl complexes, resulting in a loss of intracellular iron and the inhibition of mitochondrial respiration and DNA synthesis in the tumor cells [92]. NO can modulate tumor metabolism by inhibiting respiration, alterations in mitochondrial mass, inhibition of bioenergetic enzymes, and the stimulation of secondary signaling pathways [93]. NO inhibits DNA synthesis by inhibiting the ribonucleotide reductase, which is a rate-limiting enzyme in DNA synthesis [94]. Long-lasting levels of NO promote apoptosis by activating the caspase family proteases [85], promoting p53 expression [95], and promoting the expression of pro-apoptotic proteins including the Bcl-2 family proteins [96]. NO and reactive oxygen species are also capable of killing or sensitizing a range of different cancer cell types to other therapies, including melanoma [97], ovarian cancer [98], bladder cancer [59], and others (Table 2). The genetic deletion of iNOS has been shown to promote intestinal tumorigenesis [99] and lymphoma and sarcoma developments [100]. NOS2 and p53 knockout mice developed lymphomas and sarcomas faster and had a lower apoptotic index, increased proliferation index, and a decreased expression of death receptor ligands [100]. Confirming this finding, NO-releasing aspirin was shown to prevent intestinal tumor growth and development in mice models (*p* < 0.001) [101]. The transfection of iNOS-expressing constructs into melanoma cells has also been shown to inhibit tumor growth and metastasis [97,102,103]. Based on this information, the evidence that has been reported and discussed in the review strongly suggests that NO is directly involved in either the progression or inhibition of cancer, based on the levels and the cancer type.

### 2.3. Role in Apoptosis

The role of NO in apoptosis is complex and can either promote or inhibit apoptosis, depending on the rate of production and the interaction with other molecules. Long-lasting production of NO results in the activation of the caspase family proteases via the release of mitochondrial cytochrome c into the cytosol, up-regulation of p53, and regulation of apoptotic proteins, such as the Bcl-2 family [104]. Conversely, low levels of NO have been shown to inhibit apoptosis by activating protective proteins or inhibiting apoptotic effector proteins [104].

#### 2.3.1. As a Pro-Apoptotic Regulator

NO can promote apoptosis in various cell types including macrophages [105], thymocytes [106], neurons [107], and tumor cells [108] and can sensitize several cancers to apoptosis. For example, IFN-γ and other proinflammatory cytokines stimulate the induction of iNOS and the production of NO, which sensitize Fas-resistant human ovarian carcinoma cell lines to Fas-mediated apoptosis by upregulating the expression of the Fas receptor in the cell [109]. NO inhibits the transcription-resistant factor YY1, which results in the induction of the tumor expression of the proteins, Raf Kinase Inhibitor Protein (RKIP) and PTEN, the inhibition of the pro-survival Nuclear Factor kappa-light-chain-enhancer of activated B cells (NF-kB) and AKT pathways, and the upregulation of Fas and Death Receptor 5 (DR5) expression on tumor cells, thus reversing resistance [56]. NO has been found to sensitize prostate carcinoma cell lines to TRAIL-mediated apoptosis by downregulating NF-kB activity and the expression of the anti-apoptotic Bcl-2 related gene (*Bcl-xL*) [110]. NO sensitizes neuroblastoma cells to apoptosis by ionizing radiation by inhibiting Mdm2-mediated nuclear export of p53, thus promoting the nuclear retention and activation of p53 [111]. NO, produced by the oxidation of the NO-donor, JS-K, increased cell apoptosis of bladder cancer cells in a concentration-dependent manner and time-dependent manner by increasing ROS levels [59]. Nitrites, which were generated from the oxidation of JS-K-released NO, contributed to the induced apopotsis through the ROS-related pathway [59].

#### 2.3.2. As an Anti-Apoptotic Regulator

Long-lasting levels of NO can prevent apoptosis in hepatocytes by inhibiting seven caspase family proteases via S-nitrosylation of the catalytic cysteine residue [85]. NO has also been shown to prevent H_2_O_2_-induced apoptosis in human neuroblastoma cells by inhibiting the proteolytic activation of caspase-3 and the mitochondrial cytochrome c release [112]. This activity of NO suppresses apoptosis signaling that mediates the interaction between 14-3B and Bad phosphorylation via PKG/PI3k/Akt, resulting in the inhibition of apoptosis [112].

### 2.4. Role as an Immune Mediator

NO is involved in various innate and adaptive immune processes. NO is produced by activated macrophages and is involved in macrophage-induced cytotoxicity [91]. The expression of iNOS in macrophages is transcriptionally controlled by cytokines and bacterial pathogens through the activation of nuclear transcription factors, such as NF-κB [113]. NO acts as a regulatory molecule for the growth and death of many immune cells, including macrophages, T-lymphocytes, antigen-presenting cells, mast cells, neutrophils, and natural killer cells [114].

In innate immunity, NO exerts its antimicrobial effect and acts as a toxic defense molecule against infectious organisms [115,116,117]. NO antimicrobial polymers have recently been developed and shown to be effective for various antimicrobial applications, including the eradication of biofilms and therapy for *Staphylococcus aureus* and other antibiotic-resistant skin infections [118,119,120].

Although the production of NO by macrophages is believed to have evolved for its antimicrobial effects, NO has also been shown to have various immunosuppressive effects. For example, iNOS knockout mice exert an enhanced Th1 immune response to *Leishmania* infection, demonstrating its role as an immunosuppressor molecule [121]. Additionally, mice with the inactivation of iNOS are more susceptible to autoimmune encephalomyelitis [122]. NO suppresses IL-4 and antibody responses to *Salmonella typhimurium* infections in mice [123]. NO exerts an immunosuppressor effect in mice harboring other infections as well, including *Echinococcus multiocularis* [124], *Trypanosoma cruzi* [125], and *Plasmodium vinckei* [126].

NO also plays a role in tumor-induced immunosuppression. In a rat model of colon cancer, an inhibitor of NO was shown to restore lymphocyte proliferation, revealing a role of NO in the suppression of T-lymphocytes in cancer [127]. NO was later found to suppress T-lymphocyte proliferation via the suppression of Stat5 phosphorylation [128]. Superoxide may enhance T-cell mediated immunity by inhibiting the immunosuppressive activity of NO [129]. The NO scavenger carboxy-potassium salt reversed the immunosuppression by NO in a murine model of melanoma and enhanced the proliferative capacity and function of cytotoxic lymphocytes, resulting in the suppression of tumor growth [130]. The nitric oxide carrier, S-nitroglutathione, has been shown to reduce the immunosuppression in epithelial ovarian cancer by reducing the immunosuppressive myeloid-derived suppressor cells and enhancing the cytotoxic T-cell activity [131].

NO can also have either a proinflammatory or an anti-inflammatory effect, depending on the type of inflammation and the concentration of NO [132,133,134,135,136]. NO in inflamed tissues is a key factor in promoting carcinogenesis and tumorigenesis [90,137,138]. NO reacts with the superoxide anion O_2_^−^ to produce peroxynitrite (ONOO^−^), which may be cytotoxic, or can decompose to form HO^−^, which is a toxic radical [139]. NO can also be oxidized to NO_2_^−^, which induces DNA damage [140]. NO itself can also induce mutations in human lymphoblastoid cells [141] and induce DNA damage by causing strand breaks and deaminating nucleobases, including 5-methylcytosine [141,142].

Because of the many conflicting results of studies on NO in cancer, NO is now believed to play a complex role in cancer that is dependent on many factors, including the source of the NO, the activity of NOSs, the tumor microenvironment, the target cell type, and the concentration and duration of exposure [67,85,143]. While some NO-based cancer therapies are in development, they are still in the primitive stages and have room for improvement. Understanding the exact roles of NO in cancer processes and what other factors influence its effects is important in the further development and success of NO-based cancer therapies.

## 3. Nitric Oxide in Overcoming Immune and Chemo Resistance

Several studies have shown that NO plays a role in sensitizing tumor cells to chemotherapies and apoptosis and may be a potential agent to aid in overcoming drug resistance. For example, NO was found to sensitize human ovarian carcinoma cell lines to Fas-mediated apoptosis [110] and prostate carcinoma cell lines to TRAIL-mediated apoptosis [111]. NO was found to promote the activation of the p53 tumor suppressor and sensitize neuroblastoma cells to apoptosis by ionizing radiation [112]. iNOS induction has also been shown to enhance the toxicity of cisplatin in vivo for prostate and colon cancer cell lines, revealing the role of NO as a sensitizer for cisplatin chemotherapy [144].

Several mechanisms have been proposed for the role of NO in the reversal of immune resistance. NO is believed to dysregulate the NF-kB/SNAIL/YY1/RKIP/PTEN loop in tumor cells by repressing SNAIL, YY1, the prosurvival NF-kB pathway, and the anti-apoptotic AKT pathway [56]. NO also induces RKIP, PTEN, Fas, and DR5 expressions, leading to the sensitization of tumor cells to FasL, TRAIL, and chemotherapeutic-induced apoptosis [56]. Because YY1 inhibition by NO leads to the sensitization of cancer cells to both chemotherapy and immunotherapy [145], and YY1 mediates the expression of PD-L1 in tumor cells [35], NO may play a role in sensitizing tumor cells to anti-PD1/anti-PD-L1 therapy. The treatment of tumor cells with a NO donor resulted in the inhibition of YY1 DNA-binding activities by S-nitrosylating the cysteine residues involved in DNA binding [146]. The nitric oxide donor, RRx-001, combined with the anti-PD-L1 checkpoint inhibitor increased the complete response rate in a preclinical mouse model of myeloma [147]. Thus, NO may play a role in the reversal of immune resistance through the inhibition of YY1 (Figure 1).

Another mechanism by which NO prevents immune resistance is through regulating the hypoxia-induced immune escape. A study found that hypoxia contributes to the tumor immune escape by increasing the tumor cell’s expression of the metalloproteinase ADAM10 in a hypoxia-inducible factor-1 (HIF-1α)-dependent manner, leading to the increased expression of PD-L1 on tumor cells [148]. This study also found that NO/cGMP signaling inhibits these hypoxia-induced malignant phenotypes by interfering with HIF-1α accumulation via a mechanism involving calpain. HIF-1α is a transcription factor, and its high expression is associated with poor outcomes in most malignancies and tumor resistance to therapy [149]. In vivo experiments have shown that, under hypoxic conditions, YY1 contributes to the accumulation of HIF-1α and the expression of its target genes by stabilizing HIF-1α [150]. This study also inhibited YY1 with siRNA and found that this inhibition disrupted HIF-1α stabilization, decreased the expression of its target genes, and significantly suppressed the growth of metastatic cancer cells. Since it has been found that NO inhibits YY1 activity [58], NO may prevent hypoxia-induced immune escape through inhibiting YY1 and, thus, the accumulation of HIF-1α (Figure 2). A synthetic furoxan-based NO-releasing derivative of bifendate has been shown to have anticancer effects in multi-drug-resistant cells through several mechanisms, including the inhibition of HIF-1α expression [151]. NO-reversal of HIF-1α stabilization can also lead to increased radiosensitivity for prostate cancer [152].

NO-induced sensitization to chemotherapy was also found to be caused by the inhibition of NF-kB and Snail, the expression of the downstream anti-apoptotic genes of NF-kB, and the activation of the tumor suppressor gene Raf-1 Kinase Inhibitor Protein (RKIP) and PTEN expressions [153]. The NF-kB inhibition was the result of S-nitrosylation of the p50 and p65 polypeptide chains of NF-kB. The inhibition of NF-kB is also a mechanism through which YY1 expression is inhibited. The increased expression of PTEN by NO is a result of the inhibition of YY1. The induction of PTEN inhibits the PI3K/AKT pathway, thus contributing to the sensitization to chemotherapy [153]. However, these findings were based on a study done on prostate cancer cells [145]. Another study done on ovarian cancer cell lines found that NO donors do not affect the expression of PTEN; however, NO donors can still enhance cisplatin-induced cytotoxicity, although to a minor extent [154]. This study also found that the action of NO donors varies among different cancers and is strongest in low aggressive and cisplatin-sensitive cancer cells [154]. The sensitization was seen only at high NO concentrations and long exposure times [155]. The sensitization was due to the inhibition of STAT3 and AKT [156] and the increase in late apoptosis/necrosis of cancer cells, which is believed to be caused by a depletion of cellular ATP [154]. The concentration of NO is very important on the effect it can have in ovarian cancer cell lines because, at low concentrations, it can increase glycolysis and cell proliferation [156]. Thus, it is evident that NO can sensitize different types of cancer cells to apoptosis through various mechanisms, although the extent may be different with varying cancer types, and the effect may differ at varying concentrations. The following section will discuss the applications of these findings for the development of various therapies.

## 4. NO Induction and NO Donors in Cancer Therapy

Several NO-based therapies have been tested in nonclinical and clinical studies. In various solid tumor types, the nitric oxide donor, RRx-001, has been clinically shown to induce NO production under hypoxic conditions [157], exhibit synergistic cancer cell cytotoxicity when used in combination with radiation therapy [158], and protect against cisplatin-induced toxicities [159]. These effects were seen in the following tumor types: colorectal, hepatocellular carcinoma, melanoma, head and neck, pancreatic, ovarian, cholangiocarcinoma, lung, and oligodendroglioma [157]. Additionally, NO-donating drugs, such as aspirin, have been shown to suppress tumorigenesis of several types of tumors, including lung cancer in vitro and in vivo through modulating EGFR signaling [160].

Because of the systemic toxicities that NO donors may cause, such as cytokine release syndrome, there remains a need for the development of new NO donors that have a localized effect, in order to prevent systemic effects. Recent approaches have focused on controlling the release and delivery of nitric oxide to inhibit cancer cell growth, such as with the use of nanoparticles [161,162,163,164,165]. These approaches have shown promising success, with NO improving platinum cancer therapy [163,166], enhancing cytotoxicity for synergistic treatments, achieving multidrug resistance reversal in mice models [164], and improving chemotherapy and preventing side effects of traditional chemotherapy [167].

Targeting NO synthases for gene therapies is another approach to regulate NO production. A study found that NO deficiency caused by a decrease in eNOS activity was an early event in breast cancer pulmonary metastasis [168]. Thus, targeting eNOS for activation in early breast cancer may be a preventative approach for preventing pulmonary metastasis. Other studies have developed gene therapies by which the inducible NO synthase gene is transfected into cancer cells. For example, the adenovirus-mediated transfection of iNOS into bladder carcinoma cells resulted in an increase in NO concentration, P53 expression, and apoptosis [169]. Additionally, the cationic liposome-mediated transfection of iNOS into lung cancer enhanced various cisplatin-induced antitumor effects [170]. However, it is important to note that NO can have a procancer role in several types of cancers. For example, iNOS transfection into triple-negative breast cancer was shown to increase EGFR activation, proinflammatory cytokines, and cell invasion [171]. The transfection of iNOS in pancreatic cancer also allowed the tumor to escape the immune response [172]. NO has also been found to contribute to metastatic melanoma development [173] and human oral cancer [174]. The inhibition of iNOS has been identified as a therapeutic strategy to target these cancers. Thus, the effects of the transfection of NO synthases in cancer are highly dependent on the cancer type. A summary of recent NO-based cancer therapies is shown in Table 3.

## 5. Future Perspectives and Conclusions

It is becoming increasingly clear that NO is involved in cancer immunity and can enhance the effectiveness of immunotherapies, chemotherapies, and radiotherapies. However, the effects of NO in cancer are highly dependent on the cancer cell type, the source of the NO, the localized concentration of NO, and changes in the tumor microenvironment. NO and NO donors have been clinically shown to have antitumor effects in certain cancers (Table 3). However, NO has also been shown to have various tumor-enhancing effects in triple-negative breast cancer [171], pancreatic cancer [172], melanoma [173], and oral cancer [174]. Although there have been several studies that have shown that iNOS can contribute to the growth of metastatic melanoma through modulating the tumor microenvironment [173], a recent study has shown that nonthermal plasma can induce immunogenic cell death of melanoma cancer cells [191]. This study was done in a vaccination assay in vivo using a dielectric barrier discharge system, which generated short-lived reactive oxygen species, including nitric oxide, and contributed to an anticancer response [191]. Therefore, the source and delivery of the NO to a particular cancer can make an impact on what type of effect it can have. Another study showed that equal concentrations of different NO donors can release substantially different amounts of NO over time and have different stabilities in vitro [200]. The study used NO-sensitive microsensors and found that donors such as NOC-5 and PAPA-NONOate decay a lot quicker than SNP and GSNO [200]. This study is very important because it addresses the differences between NO donors, which is essential in the application of these NO donors for cancer therapy. Growing evidence also supports that NO has an antitumor effect only at high concentrations and can have the opposite effect at low concentrations [201]. Therefore, the effects of NO-based cancer therapies likely depend on the type of NO donor, the concentration of the NO donor, and the concentration of NO that is generated.

Many recent studies have shown how NO delivery systems using nanoparticles, near-infrared lasers, high-intensity ultrasound, and photodynamic systems can have anticancer and/or immunotherapy-boosting effects (Table 3). Some of the mechanisms by which high doses of NO can have immunotherapy-boosting effects are through inhibiting YY1, HIF-1α, NF-kB and Snail, and activating the expressions of RKIP and PTEN [153]. A synthetic nitric oxide-releasing derivative of bifendate was also shown to have anticancer effects against multi-drug-resistant tumor cells via the inhibition of HIF-1α and protein kinase B (AKT), extracellular signal-regulated kinases (ERK), and NF-kB and the activation of tyrosine nitration and apoptosis [151]. This study highlights the ability of NO-releasing agents to exhibit antitumor activities and sensitize multi-drug-resistant cancer cells to apoptosis. Since drug resistance to nitric oxide donors is difficult to achieve [151], this study provides evidence for the promising potential of NO donors to reverse drug resistance.

NO is a good candidate for cancer therapies because it has pleiotropic effects against tumor cells [143]. It can sensitize tumor cells to other therapies by inducing apoptosis, along with inhibiting cell proliferation, the EMT phenotype, metastasis, and the immune resistance of cancer cells. NO donors have been shown to have synergistic effects in combination with cancer therapies, such as photodynamic therapy, chemotherapy, radiotherapy, and immunotherapy. Thus, the development and use of NO donors in clinical studies should be continued to be assessed based on their promising potential. Beyond the NO donors that have been developed already, nitroglycerin may be a promising anticancer NO donor. Nitroglycerin, a generator of NO, has been clinically used for more than 100 years for various medical conditions, mostly because of its vasodilatory effects. It has been used as a treatment for conditions, such as hypertension, angina pectoris, and congestive heart failure, and prophylaxis [202]. It has been recently identified as a potential anticancer drug because of its low systemic toxicity and its ability to reduce the levels of HIF-1α in hypoxic tumors [202]. There is some evidence that it can have a pro-apoptotic effect in prostate cancer; however, there are mixed results for non-small-cell lung cancer [202]. Nitroglycerin, along with other NO donors, should continually be assessed for their potential as anticancer drugs, alone and in combination with other therapies. While experimental studies show synergy between NO donors and therapeutic agents, it may be useful to develop conjugates of NO donors with other targeting drugs, such as antibodies, to ensure that both agents simultaneously reach the tumor cells and prevent systemic toxicities. In the combination treatments, it is suggested that subtoxic doses of the NO donor and the targeted drug are used to achieve a synergistic antitumor effect with minimal toxicity.

In order to develop more effective NO donors for cancer therapy, some basic questions must be answered. Does the donor specifically target cancer cells and prevent systemic toxicities? What specific concentration of the NO donor exhibits maximum effectiveness in a particular cancer? Does the donor consistently exhibit antitumor effects in vivo? Can the donor work synergistically with other types of cancer therapies, such as chemotherapy, radiotherapy, and immunotherapy?

Overall, while the antitumorigenic and pleiotropic roles of NO and NO donors in cancer have been documented in several studies, caution must be taken for the clinical use to minimize toxic effects. However, the findings of this study suggest promising potential for the development of combination therapies in which NO donors are used in combination with chemotherapies, radiotherapies, and immunotherapies.

## Figures and Tables

**Figure 1 antioxidants-08-00407-f001:**
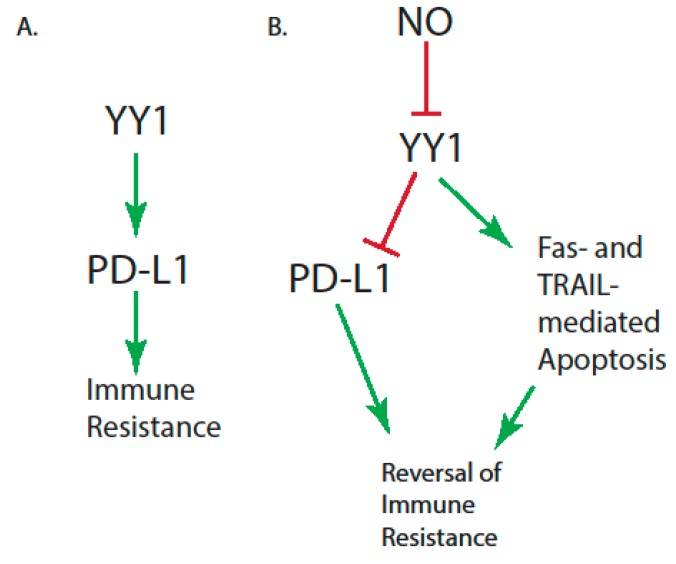
NO, YY1, and PD-L1 in the reversal of immune resistance. (**A**) YY1 mediates the expression of PD-L1 in tumor cells, leading to CTL-mediated immune resistance. (**B**) NO inhibits the activity of YY1, leading to, on the one hand, the inhibition of PD-L1 expression and DR5 expression and, on the other hand, the upregulation of Fas. Both of these effects of NO sensitize tumor cells to CTL-mediated cytotoxicity and tumor regression.

**Figure 2 antioxidants-08-00407-f002:**
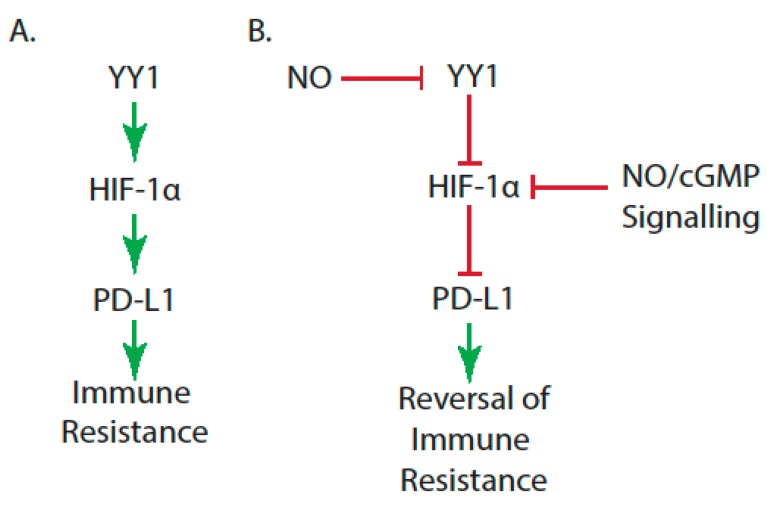
NO, HIF-1α, and YY1 in the reversal of immune resistance in hypoxic conditions. (**A**) YY1 contributes to the stability and accumulation of HIF-1α, leading to the upregulation of PD-L1 and tumor cell immune resistance. (**B**) NO inhibits YY1, thus reversing HIF-1α accumulation, the expression of PD-L1, and immune resistance. NO/cGMP signaling also inhibits HIF-1α accumulation in a mechanism involving calpain, thus reversing the expression of PD-L1 and immune resistance.

**Table 1 antioxidants-08-00407-t001:** Examples of Immune Resistance Mechanisms.

Immune Resistance Mechanism	Reference
Absence of good bacteria in the gut including *Akkermansia muciniphila*	[32]
High levels of Yin-Yang 1 (YY1), which modulate programmed death ligand 1 (PD-L1) expression	[35,36]
Absence of tumor antigens	[37]
Downregulation or mutation of MHCs and decreased antigen presentation	[38]
T-cell exhaustion mediated by up-regulation of PD-L1 and cytotoxic T-lymphocyte-associated protein 4 (CTLA-4) expression	[39,40]
Loss of Phosphatase and Tensin Homolog (PTEN) expression and activation of the PI3K-AKT pathway	[41]
High levels of Interferon gamma (IFN-γ), which drives expression of PD-L1	[42]
Lack of T-cells with tumor antigen-specific receptors	[43]
Presence of inhibitory receptors on immune cells (V-domain Immunoglobulin Suppressor of T-cell Activation (VISTA,) Lymphocyte Activating Gene 3 (LAG-3,) and T-cell Immunoglobulin and Mucin Protein 3 (TIM-3))	[44]
Immunosuppression caused by: The increased production of immature myeloid cells in cancer patients Signal Transducer and Activator of Transcription 3 (STAT3) activity Snail during cancer metastasis Dendritic cell dysfunction	[45,46,47,48], respectively

**Table 2 antioxidants-08-00407-t002:** Examples of Sensitizing Agents that Induce Apoptosis of Tumor Cells.

Sensitizing Factor	Type of Sensitization	Type of Tumor Cell	Reference
Nitric Oxide Donors (Inhibit YY1 and NF-kB and upregulate DR5)	Tumor Necrosis Factor-related apoptosis-inducing ligand (TRAIL)-mediated apoptosis	Prostate carcinoma cells	[56,58]
Reactive oxygen species	JS-K-induced cell apoptosis	Bladder cancer cells	[59]
Melatonin	Reactive oxygen species-induced apoptosis	HeLa cervical cancer cells	[60]
Cetuximab (EGFR antibody)	Reactive oxygen species-induced apoptosis	Head and neck squamous cell carcinoma	[61]
Biguanides and Rotenone (superoxide inducers)	ABT-737-induced apoptosis	Leukemia cells	[62]
AZD1208 (Pan-Pim kinase inhibitor) and Topoisomerase 2 inhibitor (chemotherapy drug)	Reactive oxygen species-induced apoptosis	Acute Myeloid leukemia	[63]
Mitochondria targeting molecules that shift cells from Glucose to Fructose metabolism	Rotenone and reactive oxygen species-induced apoptosis	Jurkat leukemia cells	[64]

**Table 3 antioxidants-08-00407-t003:** NO-based therapies for various cancers and their effects.

NO-Dependent Therapies	Antitumor Effect	Reference
NO production by tumor-infiltrating myeloid cells	Important for adoptively transferred CD8+ cytotoxic T cells to destroy tumors	[175]
RRx-001 (NO donor)	Cancer cell cytotoxicity and protection of cisplatin-induced toxicities Induction of apoptosis and reversal of drug resistance in multiple myeloma cells	[157,159,176]
NO-donating β-elemene hybrids	Inhibited tumor growth in liver tumors	[177]
Type I IFNs, IFN-a and IFN-b	Synergized with Toll-like Receptor (TLR) agonists for transcription of iNOS mRNA and secretion of NO and inhibited cancer cell growth of lewis lung carcinoma	[178]
NO-donating aspirin	Suppressed tumorigenesis in vitro and in vivo through modulation of the Epidermal Growth Factor Receptor (EGFR) signaling pathway in lung tumors	[160]
Coupling of photodynamic therapy with photocontrolled release of NO	Synergistic therapeutic effects via various mechanisms	[179]
Increase in NOS expression and nitric oxide levels triggered by silver nanoparticles	Induced apoptosis of pancreatic ductal adenocarcinoma	[180]
NO generators nitroglycerin, hydroxyurea, and l-arginine	Improved the therapeutic effects of the polymer-conjugated pirarubicin and increased delivery of nanomedicine to solid tumor models in end-stage breast cancer	[181]
NO-donor DETA/NO combined with clopidogrel	Improved vasoprotective and antiplatelet activity and reduced lung metastatic foci formation in metastatic mammary gland cancer	[182]
Intracellular enzyme-triggered NO-generator	Tumor cytoplasm-specific disruption and localized doxorubicin rapid drug release, increased apoptosis by NO	[183]
Endogenous production of NO by chloroquine and bortezomib	Enhanced doxorubicin’s cytotoxicity by inducing C/EBP-β LIP induction and inhibiting P-glycoprotein activity in triple-negative breast cancer	[184]
NO release into tumor cells by iNOS within tumor-infiltrating macrophages	Intracellular accumulation of toxic secondary oxidants, such as peroxynitrate, increased apoptosis through activation of the mitochondrial pathway	[185]
JS-K (NO donor)	Induced autophagy and inhibited tumor growth of ovarian cancer	[165]
N-heterocyclic carbene-based NO donors delivered by high-intensity ultrasound	High heat and tumor growth inhibition	[186]
Near-infrared laser-controlled NO release of sodium nitroprusside-doped Prussian blue nanoparticle	Photothermal effect in vivo and in vitro of breast cancer cells	[187]
Near-infrared laser-triggered NO nanogenerators	Reversal of multidrug resistance (MDR) via inhibition of the expression of P-glycol in an in vivo humanized MDR cancer model	[164]
NO-releasing selective estrogen receptor modulators	Anti-proliferative effect in breast cancer and melanoma cells	[188]
Graphene oxide platinum nanoparticle nanocomposites	Increased pro-apoptotic genes and decreased anti-apoptotic genes in prostate cancer	[189]
S-nitrosothiols and H2S donors	Effective in killing cancer cells but not normal cells	[190]
Nonthermal plasma delivery of NO	Immunogenic cell death of melanoma cells	[191]
Anti-CD24 Antibody-NO conjugate	Induced apoptosis of tumor cells and suppressed tumor growth in vitro and in vivo in hepatic carcinoma	[192]
NO-donor and Parp inhibitor combination	Sensitized cells to ionizing radiation treatment in BRCA1/2-proficient tumors	[193]
NO production from a combination of 5-aminosalicylic acid and hyperthermia	Induced apoptotic cell death of oral squamous cell carcinoma	[194]
Switchable NO-releasing nanoparticle activated by near-infrared radiation	Induced tumor vascular permeability, improved drug accumulation, blocks metastasis, and directly kills cancer cells	[195]
Nanoparticles loaded with doxorubicin and the NO-donor, S-nitrosothiol	Activated endogenous matrix metalloproteinases, which degrade collagen in the tumor extracellular matrix	[196]
pH-sensitive liposomal polymer that delivers the NO- donor DEANONOate and paclitaxel into cancer cells	Reversed a negative charge to a positive charge in the tumor microenvironment leading to the improvement of cell uptake of paclitaxel and the release of DETANONOate in the lysosome of multi-drug-resistant cancer cells	[197]
H2S donors	Increases iNOS and NO and restricts tumor development of hepatocellular carcinoma	[198]
Combination of a NO donor and photodynamic therapy	Increased cytotoxic effect in vitro and in vivo	[199]

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
