# Peer review of "Nitric Oxide-Mediated Enhancement and Reversal of Resistance of Anticancer Therapies"

_antioxidants, 2019, doi:10.3390/antiox8090407_

Round 1

Reviewer 1 Report

Dear Authors

The paper is well organized and complete in each part. The topic is interesting for a broad range of researchers, some minor error are present.

The paper deserves publication after a minor revision.

Best Regards

Author Response

Review 1: No specific revisions

Reviewer 2 Report

The role of NO donors in combination cancer therapy is underappreciated so review of this topic is timely and important. In general, the review is easy to follow, but some statements are overly optimistic or vague.

Specific comments:

The text does not convince the reviewer that NO is mechanistically and directly involved in cancer progression or inhibition and not just correlated; is NO a cause or effect of cancer progression?

Abstract – The efficacy of immune therapies is over-stated as the majority of patients do not respond

Lines 40-41 – define “significant clinical responses”

Line 81 – define “significant antitumor activity”

Line 92 states that “most patients who respond to current therapies develop long-lasting responses.” This statement needs to be revised as the average survival time for many targeted therapies is < 1 year (targeted melanoma drugs are classic examples).

Line 152 – “the following” makes the sentence awkward

Line 171-2: how does increased expression or activity of NO synthases suggest a role in progression of cancer?

Line 207 – is lack of iNOS the result of genetic deletion or pharmacological inhibition?

Line 292 – “it has come to the conclusion” needs to be revised; what is “it”?

Line 336 – how did “study” inhibit YY1?

Please add some additional discussion regarding potential toxicities of NO donors, including cytokine release syndrome, and how to avoid these toxicities.

The authors describe the complexity of NO in cancer, apoptosis and immunity. Can, and if so, how can NO donors be used most effectively for cancer treatment? What basic questions need to be answered in order to use these compounds effectively?

Additional analysis of the studies cited should be included (not just what was done), including strengths and weaknesses of the most important studies, and future directions for the field should be provided.

Author Response

Reviewer 2 suggestions and changes: 
1. Insert on line 213:  "Based on this information, the evidence that has been reported and discussed in the review strongly suggests that NO is directly involved in either the progression or inhibition of cancer, based on the levels and the cancer type." 
2. Line 8-10 “In the last decade, immune therapies against human cancers have emerged as a very effective therapeutic strategy in the treatment of various cancers, many of which are resistant to current therapies.” CHANGE TO “In the last decade, immune therapies against human cancers have emerged as a very effective therapeutic strategy in the treatment of various cancers, some of which are resistant to current therapies.”
3. Lines 40-41: “Chemotherapy has led to significant clinical responses in many cancers such as…” CHANGE TO “Chemotherapy has led to significant clinical responses, including increased cytotoxicity and prolonged overall and progression-free survivals, in many cancers such as…”
4. Line 86: “Small molecule inhibitors, such as these, have shown significant antitumor activity in various types of cancer...” CHANGE TO “Small molecule inhibitors, such as these, have shown significant antitumor effects, including the increased apoptosis of tumor cells, and improved clinical outcomes in various types of cancer.”
5. Line 92: “Although most patients who respond to current therapies develop long-lasting responses, a subset is intrinsically resistant and many of the initially responsive patients relapse and acquire resistance to further therapy…” CHANGE TO “Although many patients initially respond to current therapies, the response of many patients is not long-lasting. Additionally, a subset of patients is intrinsically resistant and many of the initially responsive patients relapse and acquire resistance to further therapy.  
6. Line 152: “The following will discuss” CHANGE TO “In the following sections, we will discuss the current progress of research…”
7. Line 171-172: “Several studies have detected elevated expression and/or activity of NO synthases in cells from various human cancers, suggesting a role of NO in the progression of cancer” CHANGE TO “Several studies have detected elevated expression and/or activity of NO synthases in cells from various human cancers, suggesting a correlative relationship between NO and the progression of cancer.” 
- (mechanisms by which NO has been shown to contribute to the progression of cancers is discussed in the following paragraph)
8. Line 210: “A lack of iNOS has been shown to promote intestinal tumorigenesis (99) and lymphoma and sarcoma developments (100)” CHANGE TO “The genetic deletion of iNOS has been shown to promote intestinal tumorigenesis (99) and lymphoma and sarcoma developments (100).”
9. Line 297: “Due to the many conflicting results of studies, it has come to the conclusion that NO plays a very complex role in cancer that is dependent on many other factors.” CHANGE TO “Due to the many conflicting results of studies on NO in cancer, NO is now believed to play a complex role in cancer that is dependent on many factors.”
10. Line 341: “This study also inhibited YY1 and found that this inhibition disrupted HIF-1α stabilization…” CHANGE TO “This study also inhibited YY1 with siRNA and found that this inhibition disrupted HIF-1α stabilization…”
11. Line 380: “There remains a need for the development of new NO donors that have a localized effect, in order to prevent systemic effects.” CHANGE TO “Due to the systemic toxicities that NO donors may cause, such as cytokine release syndrome, there remains a need for the development of new NO donors that have a localized effect. Recent approaches…” 
12. Insert after line 445: 
a. In order to develop more effective NO donors for cancer therapy, some basic questions must be answered. Does the donor specifically target cancer cells and prevent systemic toxicities? What specific concentration of the NO donor exhibits maximum effectiveness in a particular cancer? Does the donor consistently exhibit antitumor effects in vivo? Can the donor work synergistically with other types of cancer therapies, such as chemotherapy, radiotherapy, and immunotherapy? 
13. Line 419: Insert after “Another study showed that equal concentrations of different NO donors can release substantially different amounts of NO over time and have different stabilities in vitro (201)”: 
ADD: “The study used NO-sensitive microsensors and found that donors such as NOC-5 and PAPA-NONOate decay a lot quicker than SNP and GSNO (201). This study is very important because it addresses the differences between NO donors, which is essential in the application of these NO donors for cancer therapy.” 
Line 432: Insert after “A synthetic nitric oxide-releasing derivative of bifendate was also shown to have anticancer effects against multi-drug resistant tumor cells via the inhibition of HIF-1α and protein kinase B (AKT), extracellular signal-regulated kinases (ERK), and NF-kB and the activation of tyrosine nitration and apoptosis (152).”:
ADD: “This study highlights the ability of NO-releasing agents to exhibit antitumor activities and sensitize multidrug resistant cancer cells to apoptosis. Since drug resistance to nitric oxide donors is difficult to achieve (152), this study provides evidence for the promising potential of NO donors to reverse drug resistance.” 
Future directions are briefly discussed in the last section; however, the inserted paragraph from previous suggestion (after line 445) addresses how to develop more effective NO donors in the future. 

Reviewer 3 Report

In this review, Hays and Bonavida put their attention to the treatment of resistant tumors using nitric oxide donors. In particular,r their focus has been on the transcription factor Yin-Yang 1 and on the programmed death ligand 1.
In the introduction, they have introduced briefly cancer and conventional therapies, they have also taken into account the resistance to various therapies and mechanisms involved in the resistance, and finally how reverse the resistance mechanism. They focused principally on immune resistance mechanisms, but I think that they could speak briefly also of other mechanisms like drug efflux transporter. In the second paragraph, they have introduced nitric oxide and cancer with its dual role. In the paragraph "role as immune mediator", they have treated the role of nitric oxide in the immune system, they have treated in depth also antimicrobial and anti-inflammatory activity of nitric oxide donors. I think that in this review this is not necessary.
In the paragraph future perspectives and conclusions probably is better to stress the problems involved in the contemporary use of a nitric oxide donor and a cytotoxic drug. How could be sure that NO and anticancer drug reach the same target? That their pharmacokinetic profile is similar? How could be manage the different concentrations necessary of the nitric oxide donor and the anticancer drug chosen?
This review could be published after minor revisions.

Author Response

1. Line 113- 114: “Other factors have been shown to be involved in acquiring resistance to immunotherapies, such as the transcription factor YY1.” CHANGE TO “Other factors have been shown to be involved in acquiring resistance to immunotherapies, such as the drug efflux transporter and other membrane drug transporters which shuttle drugs across cell membranes, protecting the cell from the accumulation of toxic drugs (34). The transcription factor, YY1, has also been shown to regulate immune resistance…”
a. Insert in references: (34)
Löscher, W., & Potschka, H. (2005). Drug resistance in brain diseases and the role of drug efflux transporters. Nature Reviews Neuroscience, 6(8), 591.
All following references re-numbered
2. Line 260: Take out “The inhibition of NOS has been shown to exacerbate infection and diseases caused by many types of viruses, bacteria, and other pathogens (116). IFN-γ-induced nitric oxide has been shown to inhibit viral replication in macrophages (117).” (Keep references)
Line 282: Take out: “NO is stimulated by lipopolysaccharide and proinflammatory cytokines, including IFN-γ, IL-1, and IL-2 (132). NO prevents pathogen-permissive granulocytic inflammation during tuberculosis (133) and has cytoprotective actions in hepatic inflammation (134). The transfection of eNOS was associated with the hepatic inflammation and proinflammatory M1 and antinflammatory M2 activation of Kupffer cells (135). The effect of NO in contact hypersensitivity has been shown to be dependent on its concentration. At low concentrations, it is pro-inflammatory by inducing vasodilation and at high concentrations, it is anti-inflammatory by inducing apoptosis of inflammatory cells (136).” (Keep references)
3. New Paragraph added after line 445 addresses some questions that must be answered for future NO drug developments.
Insert after inserted paragraph after line 445: ADD “While experimental studies show synergy between NO donors and therapeutic agents, it may be useful to develop conjugates of NO donors with other targeting drugs, such as antibodies, to ensure that both agents simultaneously reach the tumor cells and prevent systemic toxicities. In the combination treatments, it is suggested that sub-toxic doses of the NO donor and the targeted drug are used to achieve a synergistic antitumor effect with minimal toxicity.” 

Round 2

Reviewer 2 Report

Authors have adequately addressed previous comments

Minor corrections are still needed:

Line 177 remove the duplicated “suggesting”

The phrases “has been shown” and “is believed to” are used too frequently in this review. When references are listed at the end of a sentence, this phrase can be eliminated. For example on line 201, change “NO has been shown to have an antitumorigenic effect” to “NO has an antitumorigenic effect.” Instead of stating “NO is believed to inhibit DNA synthesis by inhibiting the ribonucleotide reductase” (line 207), the sentence can be simplified to “NO inhibits DNA synthesis.”

Lines 229-230 should be combined with the paragraph below into a single paragraph

The sentences on lines 289-291 are almost identical and should be reduced into one sentence to reduce redundancy.

Author Response

Line 177: The duplicated “suggesting” has been removed.

Line 91-92: “Combinations of small molecule inhibitors have also been shown to sensitize radiation therapy-resistant cancer cell lines”
CHANGE TO: Combinations of small molecule inhibitors can also sensitize radiation therapy-resistant cancer cell lines.

Line 119-120: The inhibition of YY1 has been shown to sensitize tumor cells... CHANGE TO: The inhibition of YY1 sensitizes tumor cells...

Line 125-127: For example, the combined use of checkpoint inhibitors for CTLA-4 and PD-1 has been shown to improve the clinical response...

CHANGE TO: For example, the combined use of checkpoint inhibitors for CTLA-4 and PD-1 can improve the clinical response...

Line 140-142: In another study done on tumors that lack CD8+ T cells, chemotherapy was shown to enhance tumor T-cell infiltration, sensitize tumors to checkpoint inhibition therapies, and control cancer growth (55).

CHANGE TO: In another study done on tumors that lack CD8+ T cells, chemotherapy tumor T-cell infiltration, sensitized tumors to checkpoint inhibition therapies, and cancer growth (55).

Line 185-186: NO derived from eNOS has been shown to promote angiogenesis and tumor progression through the maintenance of blood flow...

CHANGE TO: NO derived from eNOS promotes angiogenesis and tumor progression through the maintenance of blood flow...

Line 189-190: NO has also been shown to act as an epigenetic regulator of oncogenesis by causing histone posttranslational...

CHANGE TO: NO acts as an epigenetic regulator of oncogenesis by causing histone posttranslational...

Line 196-197: NO has also been shown to regulate proinflammatory mediators, resulting in the oncogenic transformation of cells (90)

CHANGE TO: NO also regulates proinflammatory mediators, resulting in the oncogenic transformation of cells (90).

Line 199-200: It is believed to have a cytotoxic effect due to its ability to sequester iron into iron-nitrosyl complexes,

CHANGE TO: Its cytotoxic effect is due to its ability to sequester iron into iron-nitrosyl complexes,

Line 265: NO has been shown to suppress IL-4...

CHANGE TO: NO suppresses IL-4...

Line 201: “For instance, NO has been shown to have an antitumorigenic effect by contributing to...”

CHANGE TO: “For instance, NO has an antitumorigenic effect of contributing to...”

Line 207: NO is believed to inhibit DNA synthesis by inhibiting the ribonucleotide reductase, which is a rate-limiting enzyme in DNA synthesis (94).

CHANGE TO: NO inhibits DNA synthesis by inhibiting the ribonucleotide reductase, which is a rate-limiting enzyme in DNA synthesis (94).

Line 207-208: Long-lasting levels of NO have been shown to promote apoptosis by activating the caspase family proteases (85),

CHANGE TO: Long-lasting levels of NO promote apoptosis by activating the caspase family proteases (85),

Line 280-281: NO in inflamed tissues has been shown to be a key factor in promoting carcinogenesis and tumorigenesis

CHANGE TO: NO in inflamed tissues is a key factor in promoting carcinogenesis and tumorigenesis

Lines 229-230 have been combined with the paragraph below to form a single paragraph:
Original: NO has been shown to promote apoptosis in various cell types including macrophages (106), thymocytes (107), neurons (108), and tumor cells (109).
NO has been found to sensitize several cancers to apoptosis.
CHANGE TO:

NO can promote apoptosis in various cell types including macrophages (106), thymocytes (107), neurons (108), and tumor cells (109) and can sensitize several cancers to apoptosis.

Lines 289-291: Redundancy has been removed:
Original: Due to the many conflicting results of studies on NO in cancer, NO is now believed to play a complex role in cancer that is dependent on many factors. The effect of NO on tumor progression is now believed to be dependent on several factors, including the source...

CHANGE TO:
Due to the many conflicting results of studies on NO in cancer, NO is now believed to play a complex role in cancer that is dependent on many factors, including the source...